# Zoonotic Risks of *Proteus mirabilis*: Detection, Pathogenicity, and Antibiotic Resistance in Animals and Animal-Derived Foods

**DOI:** 10.3390/microorganisms13092060

**Published:** 2025-09-04

**Authors:** Xiao-Li Liu, Si-Yi Wu, Zhongjia Yu

**Affiliations:** School of Animal Science and Technology, Foshan University, Foshan 528225, China; a879005412@163.com (X.-L.L.); wsy1319225030@163.com (S.-Y.W.)

**Keywords:** *Proteus mirabilis*, animals, transmission, antibiotic resistance, One Health

## Abstract

*Proteus mirabilis* is a major uropathogen with growing concern over its presence in animal products and the associated zoonotic transmission risks. As a gut commensal in both humans and animals, it is increasingly detected in wild, farm, and companion animals, as well as in animal-derived foods and related environments. This review summarizes current evidence on its distribution across these sources and explores potential transmission routes to humans. Special attention is given to reported genomic similarities and shared antibiotic resistance patterns between animal and human isolates. The role of *P. mirabilis* in exacerbating intestinal inflammation further highlights its relevance beyond urinary infections. By revealing the epidemiology, pathogenic traits, and resistance profiles of animal-associated isolates, this review underscores the zoonotic potential of *P. mirabilis* and emphasizes the need for enhanced surveillance and research from a One Health perspective.

## 1. Introduction

*Proteus mirabilis*, a Gram-negative facultative anaerobic bacterium, belongs to the genus *Proteus* within the family *Morganellaceae*. Traditionally classified in the *Enterobacteriaceae* family, the taxonomy of *P. mirabilis* was reassessed with advancements in genomic analysis, leading to its reclassification under *Morganellaceae* by Adeolu et al. in 2016 based on phylogenetic studies [1]. As of 9 December 2024, the LPSN database lists *P. mirabilis* as a formally recognized species within this family, alongside other species such as *Proteus vulgaris*, *Proteus cibi*, and *Proteus faecis* (https://lpsn.dsmz.de/genus/proteus, 15 May 2025). The genus *Proteus* includes ten officially named species, with additional unnamed genomic species (genotypes 4, 5, and 6) [2,3].

Morphologically, *P. mirabilis* displays remarkable features, with a cell size ranging from 1.0 to 3.0 µm in length and 0.4 to 0.8 µm in width. The bacterium is polymorphic, appearing as short rods, spheres, and filaments, and lacks both a capsule and spore structure [4,5]. It is motile due to the presence of flagella and exhibits a characteristic swarming behavior on agar surfaces. This strong motility leads to the formation of concentric rings and visible demarcation lines, known as Dienes’ lines, which are used for strain differentiation [6,7]. The bacterium also possesses pili, enabling adhesion to epithelial cells of plants and fungi [8]. *P. mirabilis* is a major uropathogen, contributing to more than 40% of catheter-associated urinary tract infections (CAUTIs) cases [9,10,11,12]. Its virulence factors, including urease, flagella, pili, hemolysins, and metalloproteases, play crucial roles in its ability to colonize the host, damage tissues, and evade immune responses [13,14]. Although typically a commensal in the intestinal tract, recent studies have linked *P. mirabilis* to inflammatory bowel diseases such as Crohn’s disease [15]. It has been shown to exacerbate colitis by disrupting the intestinal mucus barrier and modulating host immune responses [16].

Although the epidemiology and pathogenicity of *P. mirabilis* have been widely studied, its significance as a zoonotic pathogen remains poorly documented. This review summarizes current findings on the distribution of *P. mirabilis* in wildlife, farm animals, companion animals, and associated environments and products. We also aim to elucidate the potential transmission routes of this zoonotic pathogen between humans and animals, with a focus on reported genomic similarities and shared antibiotic resistance patterns. By highlighting the zoonotic risks associated with *P. mirabilis*, this review seeks to provide a basis for future research on its pathogenic mechanisms and its impact on human and animal health.

## 2. The Detection of *P. mirabilis* in Animals

As a commensal bacterium in the gastrointestinal tract of both humans and animals, *P. mirabilis* is widely distributed across diverse animal species, including wildlife, farm animals, and companion animals. This broad host range suggests potential transmission routes between domestic and wild environments, as well as between animals and humans. As shown in Table 1, the frequent isolation of *P. mirabilis* from various animal sources underscores its widespread presence.

Wildlife has been identified as a significant reservoir for *P. mirabilis*, with the bacterium isolated from a wide range of species and their environments, underscoring its zoonotic potential. As shown in Table 1, in undisturbed environments, 17 *P. mirabilis* was isolated from 23 Egyptian vulture chicks in the Canary Islands, a single strain from 37 migratory bird feces at Dianchi Lake, China as well as one isolate was found in a fruit bat in Indonesia’s Gunung Halimun Salak National Park [17,18,19]. For wild mammals, two out of 110 wild boars (2.2%) from 19 rural and urban areas of northern Tunisia were isolated *P. mirabilis* [20]. From wildlife in a national park in Gabon, 7 isolates were from gorillas, mandrills, and African buffaloes [21]. Raptors in Catalonia, Spain, including Eurasian goshawks and barn owls, also carried this bacterium [22]. Notably, it was recovered from a juvenile sea lion in Uruguay (20%, 1/5) [23], indicating its presence in marine mammals. Beyond vertebrate hosts, ectoparasites may play a role in transmission. Ergunay et al. detected *P. mirabilis* in 48.5% (17/35) of ticks collected from various wild mammals (e.g., rhinos, giraffes, lions) and domestic cattle in Kenya [24].

In wildlife held in controlled environments such as zoos or farms, *P. mirabilis* has also been successfully isolated. Liu et al. used multiplex PCR to detect *P. mirabilis* in 100 fecal samples from giant pandas in Sichuan, China, with a detection rate of 30% (35/100) [25]. In the same facility, *P. mirabilis* was identified in kidney, liver, and urine samples from 3 out of 32 deceased red pandas, yielding an isolation rate of 9.4% [26]. Lv et al. isolated 53 *P. mirabilis* strains from farmed foxes, raccoons, and minks, as well as from their surrounding environments. Specifically, 12 isolates came from fox feces, 15 from raccoon feces, 12 from minks (feces, carcasses, throats, and anal swabs), and 14 from the feeding environment (soil and surfaces) [27]. Under controlled laboratory conditions, Yu et al. detected *P. mirabilis* in 9.5% of diarrheal rhesus macaques (7/74) and 30% of ferrets (4/12) [28]. In tree shrews, which are an emerging model in biomedical research, *P. mirabilis* was the most frequently isolated species with 34 isolates came from fecal samples [29].

**Table 1 microorganisms-13-02060-t001:** Isolation of *Proteus mirabilis* from animals and animal-derived foods.

Category	Host	Region (Year)	Isolation Rate	Virulence Gene	**Reference**
Wildlife	Egyptian vulture	Spain (2022)	17 isolates ^#^	NA	[17]
	Migratory Birds	China (2024)	2.70% (1/37)	NA	[18]
	Fruit bat	Indonesia (2021)	50% (1/2)	NA	[19]
	Wild boar	Tunisia (2021)	2% (2/110)	NA	[20]
	Western lowland gorilla	Gabon (2021)	3 isolates ^#^	NA	[21]
	Mandrill	Gabon (2021)	2 isolates ^#^	NA	[21]
	African buffalo	Gabon (2021)	2 isolates ^#^	NA	[21]
	Eurasian goshawk	Spain (2019)	1 isolate ^#^	NA	[22]
	Barn owl	Spain (2019)	1 isolate ^#^	NA	[22]
	South American sea lion	Uruguay (2022)	20% (1/5)	NA	[23]
	Tick from wildlife	Kenya (2022)	48.5% (17/35)	NA	[24]
	Panda	China (2023)	30% (30/100)	NA	[25]
	Red panda	China (2022)	9.38% (3/32)	NA	[26]
	Fox	China (2022)	12 isolates ^#^	*ureC*, *zapA*, *pmfA*, *atfA*, *mrpA*, *atfC*, *hmpA*, *rsmA*, *rsbA*, *ucaA*	[27]
	Raccoon	China (2022)	15 isolates ^#^	*ureC*, *zapA*, *pmfA*, *atfA*, *mrpA*, *atfC*, *hmpA*, *rsmA*, *rsbA*, *ucaA*	[27]
	Ferrets	China (2015)	30% (4/12)	NA	[28]
	Mink	China (2020, 2022)	24.53% (13/53), 12 isolates ^#^	*ureC*, *zapA*, *pmfA*, *atfA*, *mrpA*, *atfC*, *hmpA*, *rsmA*, *rsbA*, *ucaA*, *FliL*	[27,30]
	Rhesus Monkeys	China (2015)	9.5% (7/74)	NA	[28]
	Tree shrews	China (2020)	34 isolates ^#^	NA	[29]
Farm animals	Pig	China (2021, 2022), Rome (2021), India (2021)	5.55% (30/541)–21.43% (21/98)	*ureC*, *hpmA*, *zapA*, *pmfA*, *rsbA*, *ucaA*, *mrpA*, *atfA*, *ireA*, *ptA*	[31,32,33,34]
	Broiler	China (2020, 2022), India (2021), South Africa (2024)	5.4% (26/480)–22.5% (18/80)	*ureC*, *rsmA*, *hmpA*, *FliL*, *ireA*, *ptA*, *zapA*, *ucaA*, *pmfA*, *atfA*, *mrpA*, *hlyA*, *hpmA*	[30,31,35,36]
	Duck	Egypt (2021)	14.6% (35/240)	*atpD*, *ureC*, *rsbA*, *zapA*	[37]
	Cattle	China (2020), India (2021)	23.26% (20/86)–33.33% (20/60)	*ureC*, *zapA*, *rsmA*, *hmpA*, *mrpA*, *atfA*, *pmfA*, *FliL*, *ucaA*	[30,31]
	Sheep	India (2021)	31.91% (15/47)	NA	[31]
Companion animals	Dog	China (2020, 2022, 2023), Egypt (2022), UK (2021), Thailand (2019), Europe (2016), Portugal (2018, 2021)	11.0% (48/437)–44.4% (8/18)	*ureC*, *FliL*, *ireA*, *zapA*, *ptA*, *hpmA*, *hpmB*, *pmfA*, *rsbA*, *mrpA*, *ucaA*, *rsmA*, *atfA*	[30,38,39,40,41,42,43,44,45,46]
	Cat	UK (2021), Thailand (2020), Europe (2017), Portugal (2019, 2022)	0–2.2% (4/171)	*hmpA/hmpB*, *mrpA*, *pmfA*, *ucaA*	[38,40,41,42,45]
	Pet turtle	South Korea (2018)	28.8% (15/52)	*ureC*, *rsbA*, *zapA*, *mrpA*	[47]
Animal-derived foods	Pork	China (2022, 2023), Brazil (2021), India (2021)	14.38% (23/160)–65.61% (149/227)	*mrpA*, *pmfA*, *ucaA*, *atfA*, *hpmA*, *zapA*, *ptA*, *ireA*	[31,48,49,50]
	Beef	Brazil (2021), India (2021)	27.8% (100/360)–30.90% (17/55)	*mrpA*, *pmfA*, *ucaA*, *atfA*, *hpmA*, *zapA*, *ptA*, *ireA*	[31,48]
	Mutton	India (2021)	25.51% (25/98)	*NA*	[31]
	Chicken	China (2022, 2023), Belgium (2020), Brazil (2021), India (2021), Egypt (2023)	1.51% (1/66)–100% (200/200)	*mrpA*, *pmfA*, *ucaA*, *atfA*, *hpmA*, *zapA*, *ptA*, *ireA*	[31,48,49,50,51,52]
	Duck meat	China (2023)	67.9% (84/124)	NA	[50]
	Milk/Dairy Products	India (2021), Egypt (2023)	3.45% (2/58)–22.11% (21/95)	NA	[31,51]
Other source	Aquatic products	China (2022)	7.61% (7/92)	NA	[49]
	Vegetables	China (2023)	62.5% (5/8)	*hpmA*, *mrpA*, *ptA*, *ireA*, *zapA*, *pmfA*, *atfA*	[53]

^#^ Number of isolates reported; sample size not provided, so isolation rate could not be calculated.

Farm animals are significant reservoirs of *P. mirabilis*, raising concerns for both animal health and food safety (Table 1). Chinnam et al. reported a 15.95% (26/163) isolation rate from pig rectal swabs in Andhra Pradesh, India [31], while Qu et al. found 5.55% (30/541) in pigs from Zhejiang, China, with Jinhua showing the highest rate at 8.91% [32]. In Italy, *P. mirabilis* was also found in boar semen, negatively affecting sperm motility [33]. Beyond pigs, the bacterium was isolated from healthy chickens (21.36%, 47/220), cattle (33.33%, 20/60), and sheep (31.91%, 15/47) in Andhra Pradesh [31]. In China, broilers showed a 7.07% (50/707) isolation rate [35], while South African farms reported 5.4% (26/480) from chicken manure [36].

However, the association of *P. mirabilis* with disease in farm animals raises significant concern. In Guangxi, China, Ge et al. reported a 21.42% (21/98) isolation rate from fecal and tissue samples of diseased pigs, highlighting its potential pathogenic role [34]. Similarly, Sun et al. found isolation rates of 22.5% (18/80) in diarrheal poultry and 23.26% (20/86) in cattle in northeastern China, suggesting *P. mirabilis* may worsen animal health [30]. In Egypt, Algammal et al. detected the bacterium in 14.6% (35/240) of ducks, both healthy and diseased, indicating its diverse impact on animal health [37].

Companion animals, due to their close contact with humans, represent a potential source of zoonotic pathogens like *P. mirabilis* (Table 1). This bacterium has been increasingly detected in pets, highlighting its epidemiological importance. Marques et al. reported isolation from both humans (12.5%, 3/24) and dogs (44.4%, 8/18) in households, with no detection in cats [38]. Liu et al. found *P. mirabilis* in 31.12% (75/241) of fecal samples from household and stray dogs in Sichuan, China, with higher prevalence in stray dogs (36.17%) than household dogs (27.89%) [39]. Exotic pets also carry the bacterium; Pathirana et al. isolated it from 28.8% (15/52) of pet turtles in South Korea [47].

*P. mirabilis* is implicated in urinary tract infections (UTIs) in companion animals. Fonseca et al. found it in 22.7% (145/637) of canine urine samples but only 2.2% (4/171) in cats in the UK [40]. Similarly, Moyaert et al. reported rates of 11.0% in dogs and 1.1% in cats across Europe [41]. In Thailand, Amphaiphan et al. detected *Proteus* spp. in 13.6% of dog and 16.7% of cat urine samples [42]. Beyond UTIs, P. mirabilis was isolated from 19.35% (12/62) of dogs with diarrhea in China [43], and Sui et al. found it in dogs co-infected with canine parvovirus (47.37%) and distemper (10%) [44].

## 3. Detection of *P. mirabilis* in Animal-Derived Foods

Animal-derived foods have become a major focus of public health concern due to their contamination with *P. mirabilis*. The prevalence of *P. mirabilis* in animal products varies significantly across different countries and regions, reflecting differences in hygiene practices and environmental conditions (as shown in Table 1). In Andhra Pradesh, India, beef samples exhibited the highest contamination rate at 32.73% (17/55), while chicken and pork samples showed relatively lower detection rates of 19.49% (38/195) and 14.38% (23/160), respectively [31]. In Londrina-PR regions, Brazil, chicken meat displayed the contamination rate of chicken meat was the highest at 100% (200/200), that of beef was much lower at 27.8% (100/360) [48]. In Al Qalyubia Governorate, Egypt, chicken and milk samples had contamination rates of 1.51% (1/66) and 3.45% (2/58) [51]. In Ghent, Belgium, Yu et al. reported *P. mirabilis* in 36.25% (29/80) broiler carcasses [52], while Liu et al. reported a much higher contamination rate of 66% (66/100) in fresh chicken at Hebei, China [54]. A study by Ma et al. in wet market in Chengdu, China, isolated 89 strains of *P. mirabilis* from 347 samples of chicken, pork, and aquatic products, with an overall contamination rate of 25.65% (89/347). Among these, chicken showed the highest rate at 54.39% (62/114), followed by pork 14.18% (20/141) and aquatic products 7.61% (7/92) [49]. Lan et al. found *P. mirabilis* in 490 of 579 fresh meat samples (84.63%) from five wet markets in Zhongshan, China, with chicken 78.95% (180/228), duck 67.90% (84/124), and pork 65.61% (149/227) being the most contaminated [50]. These findings indicate that poor hygiene at poultry and meat stalls may result in significant contamination and cross-contamination, particularly affecting poultry meat. Flies associated with animal-derived food also appear to serve as potential vectors of transmission. Zaher et al. detected *P. mirabilis* from flies collected on pig carcasses [55].

Currently, *P. mirabilis* has been detected in food derived from farm animals, and its epidemiology in these animals has been well-documented. Moreover, its presence in wild boar and African buffalo, which are common sources of game meat, suggests a potential route of zoonotic transmission from wildlife to humans. However, this possibility requires further investigation.

## 4. Genomic Similarity of *P. mirabilis* Between Animal and Human

High genomic similarity has been observed between *P. mirabilis* isolates from humans, animals, and associated products and environments (as shown in Figure 1). Notably, *P. mirabilis* isolates from companion animals have been shown to share high genomic similarity with human isolates, with some strain pairs originating from individuals and pets within the same household. Marques et al. collected urine samples from 76 human patients and 107 companion animals with UTIs, and subsequently constructed a phylogenetic tree based on PFGE genotyping to compare the genetic relatedness of the isolates [45]. Among 39 clusters, 17 contained both human and animal isolates, with genomic similarity ranging from 80% to 100%. One canine isolate of *P. mirabilis* showed 100% similarity with a human isolate, while a feline isolate shared over 90% similarity with a human strain [45]. In a separate study, Marques et al. identified a human–dog pair harboring genetically related *P. mirabilis* strains, exhibiting 82.5% similarity to the animal-derived clinical strain FMV4938/07, which was isolated from a dog with a urinary tract infection [38]. One fecal *P. mirabilis* isolate from a dog in a separate household clustered with two human community-acquired UTI isolates, showing 80.9% and 88.9% genomic similarity, respectively [38]. Wang et al. isolated *P. mirabilis* (CC16012 strain) from a diarrheal dog, which was closely related to the human-derived Crl143 strain from the United States [43]. Pathirana et al. found that the *mrpA* gene sequence of *P. mirabilis* from pet turtles showed 96.4% and 94.9% similarity to human isolates from UTI and respiratory infections, respectively [47]. In experimental animals, Yu et al. isolated *P. mirabilis* from diarrheal primate feces, showing 99.6% genomic similarity to the human UTI-associated strain HI4320 [28]. Similarly, five *P. mirabilis* isolates from ducks also exhibited close phylogenetic relatedness to the HI4320 strain and other reference strains from diverse sources, based on *atpD* gene sequencing [37].

Isolates from animal-derived products have shown high genomic similarity with those from human sources. Yu et al. reported that, based on PFGE analysis, the highest similarity observed between a broiler carcass isolate and a human stool isolate was 83.3%, differing by only two bands [52]. Similarly, Sanches et al. confirmed clonal relationships exclusively between chicken-derived isolates and those causing community-acquired urinary tract infections (UTI-CA), particularly within cluster C01. Notably, two strains within this cluster, one isolated from chicken meat and the other from a community-acquired urinary tract infection (UTI) patient, exhibited 100% genomic similarity as determined by PFGE [56]. Furthermore, the *bla_NDM-1_* gene in strain JZ109 showed 100% nucleotide identity (with only a single base difference) to SGI1-1NDM, which was the first reported clinical *P. mirabilis* strain from China [49].

## 5. Pathogenicity of *P. mirabilis* in Humans and Animals

The pathogenicity of *P. mirabilis* has been reported in various tissues and organs of both humans and animals, with severe cases leading to death. One of the most notable associations of *P. mirabilis* is as a major pathogen in UTIs in both humans and companion animals [41,57]. Studies indicate that *P. mirabilis* accounts for 1–10% of all UTI cases in humans [58]. In nearly 3000 confirmed UTI cases in North America, infections caused by *P. mirabilis* represented 4% of all cases. Furthermore, the incidence of catheter-associated urinary tract infections (CAUTIs) caused by *P. mirabilis* is as high as 45% or more [59]. The incidence of *P. mirabilis*-induced UTI is significantly higher in women and the elderly population [58,59,60]. In animals, *P. mirabilis* has been linked to recurrent urinary stones in dogs with urinary system disorders, as evidenced by Song et al. [61]. Herout et al. also reported a high infection rate of *P. mirabilis* in a murine CAUTI model, further highlighting the bacterium’s role in urinary infections across species [62]. These findings highlight the role of *P. mirabilis* in UTIs in both humans and animals, with important implications for medical treatment and public health.

*P. mirabilis* has also been implicated in food poisoning incidents. Between 2016 and 2017, 3.61% of reported food poisoning cases in the Datong, China were caused by *P. mirabilis*, with symptoms such as abdominal pain, diarrhea, nausea, and dizziness [63]. Wang et al. reported a food poisoning incident in Beijing in 2018, where *P. mirabilis* contamination in braised meatballs led to illness among customers, with the bacterium detected on the hands of the chef and waitstaff [64]. Furthermore, between August and December 2018, Fan et al. isolated *P. mirabilis* from 49 out of 486 diarrheal pediatric samples, yielding a detection rate of 10.1% [65]. Zhang et al. compared the feces and inflamed colon samples from Crohn’s disease (CD) patients and healthy individuals, finding a significant increase in the abundance of *P. mirabilis* in CD patients. The experimental results showed signs of colon shortening, and liver and spleen enlargement, indicating that *P. mirabilis* plays a critical role in inducing CD inflammation [15]. Additionally, Kitamoto et al. suggested that oral inflammation could exacerbate intestinal inflammation, and the use of proton pump inhibitors may promote the proliferation of *P. mirabilis* and other microbes in the intestines, thus triggering intestinal inflammation [66].

In animals, *P. mirabilis* has been associated with various cases of gastrointestinal diseases. In 2018, a bamboo rat farm in Guangdong, China reported the deaths of 400 bamboo rats due to vomiting and diarrhea, with *P. mirabilis* identified as the causative agent [67]. Yu et al. reported similar symptoms in 74 rhesus monkeys infected with *P. mirabilis*, including diarrhea and bloody stools [28]. In a rabbit farm in Henan, China, *P. mirabilis* (strain HN001) infection resulted in lethargy, yellow watery diarrhea, and mass fatalities, accompanied by multi-organ tissue damage [68]. In Lhasa, China, a breeding farm experienced deaths of breeding rabbits due to *P. mirabilis* (strain T2018) infection, presenting with diarrhea and subsequent fatality [69]. Moreover, Dong et al. identified *P. mirabilis* (strain 17f) as the primary pathogen responsible for diarrhea in lambs in the Hotan area of Xinjiang, China [70]. These reports underscore the significant role of *P. mirabilis* as a gastrointestinal pathogen in both humans and animals, with implications for public health, food safety, and animal welfare.

In addition to affecting the urinary and gastrointestinal systems, *P. mirabilis* has been implicated in a wide range of infections across various organs in both animals and humans. In human medicine, Mistry et al. reported the isolation of *P. mirabilis* from skin abscesses with an isolation rate of 21.6%, second only to methicillin-sensitive *Staphylococcus aureus* (24.3%) [71]. In pigs, Qin et al. and Chen et al. identified *P. mirabilis* (GX-PM1 and GX-Y9251 strains) as a cause of respiratory symptoms such as fever and difficulty breathing. They also noted that P. mirabilis could cross the placental barrier, resulting in fetal death [72,73]. Similarly in animals, Li et al. found that *P. mirabilis* caused multi-organ lesions and systemic inflammation in pigs, which progressed to septicemia and death [74]. In Northern Paraná, Brazil, Sanches et al. isolated *P. mirabilis* strains (LBUEL-A33 and LBUEL-A34) from broiler chickens, where the bacterium induced caseous exudates and hemorrhaging in the subcutaneous tissue, leading to condemnation in poultry industry. Histopathological analysis revealed edema, congestion, and necrosis in the pectoral muscles, along with cellulitis and infiltration of inflammatory cells [75].

Furthermore, *P. mirabilis* has been reported to cause severe infections in other animal species. Abdollahi et al. and Ghahremani et al. were the first to document *P. mirabilis*-induced pyoderma and purulent pericarditis in sheep [76,77]. Sacristán et al. identified *P. mirabilis* as a significant causative agent of neck abscesses and bacteremia in sea lions [78]. Pattanayak et al. observed *P. mirabilis*-induced hemorrhaging in the glomeruli and localized necrosis with mononuclear cell infiltration in the kidneys of infected Indian carp [79].

## 6. Antibiotic Resistance of *P. mirabilis* from Animals and Animal-Derived Products

Antibiotic resistance in *P*. *mirabilis* from animals and food sources is a growing public health concern (Appendix A). In wildlife, isolates from fruit bats, wild boars, gorillas, mandrills, African buffaloes, and sea lions show intrinsic tetracycline resistance. Fruit bat isolates are susceptible to amoxicillin-clavulanate and cefoxitin but resistant to oxacillin. Sea lion isolates resist amoxicillin-clavulanate but are sensitive to cefovecin. For aminoglycosides, fruit bat isolates are amikacin-sensitive; wild boar isolates resist gentamicin; sea lion isolates resist streptomycin but remain gentamicin-sensitive. Sea lions also show resistance to doxycycline, trimethoprim-sulfamethoxazole, and azithromycin but remain susceptible to quinolones (ciprofloxacin, enrofloxacin) [19,20,21,22,23]. Wild raptors (Eurasian goshawks, barn owls) resist quinolones (ciprofloxacin, nalidixic acid), sulfonamides, and polymyxin but are aminoglycoside-sensitive [22]. Among farmed wildlife, *P. mirabilis* from foxes display high resistance to cefotaxime (94.4%), gentamicin (83.3%), and ampicillin (88.9%), significantly higher than raccoons and minks. Minks showed narrower resistance, remaining sensitive to cefotaxime, ceftazidime, and ofloxacin. Most isolates were imipenem-resistant (71.7%) [27].

In farmed livestock, pig isolates are multidrug-resistant (MDR), with 100% resistance to tetracyclines, ampicillin, and sulfonamides. Imipenem and kanamycin resistance reach 85.7%; gentamicin and amikacin resistance are lower (9.5%); ciprofloxacin resistance is 38.1%, fosfomycin 28.6% [34,72]. Another study found 100% pig isolates resistant to tetracyclines, chloramphenicol, and macrolides, with high ciprofloxacin resistance (76.67%) but sensitivity to meropenem. Gentamicin and amikacin resistance were 56.67% and 20%, respectively [32]. In poultry, all 50 chicken isolates in Shandong were MDR, with over 50% resistance to β-lactams (cefazolin, ceftriaxone, cefuroxime) and aminoglycosides (tobramycin, gentamicin). Resistance to ciprofloxacin, chloramphenicol, and trimethoprim-sulfamethoxazole reached 98% [35]. Broiler isolates showed 30.7% MDR, with ciprofloxacin resistance at 61.5% and gentamicin at 38.5% [36]. Duck isolates showed 100% resistance to amoxicillin, penicillin, trimethoprim-sulfamethoxazole, and doxycycline; 31.4% were extensively drug-resistant (XDR), and 8.6% pan-drug-resistant (PDR) [37]. Environmental isolates include *P. mirabilis* carrying *bla_NDM-1_* from houseflies on a sheep farm [80], and multidrug-resistant strains from captured houseflies resistant to streptomycin, trimethoprim-sulfamethoxazole, and amoxicillin [81].

In companion animals, 75 isolates from dogs in Chengdu showed 53.33% MDR; pet dogs had 75% MDR versus 25% in stray dogs. Pet dog isolates showed β-lactam resistance (e.g., cefotaxime 31.71%, ciprofloxacin 36.59%), while all were tetracycline-resistant (100%). Stray dog isolates remained largely susceptible [39]. This suggests that antibiotic use in pet clinics drives resistance.

From diseased animals, Portuguese companion animal isolates from UTIs showed moderate resistance to β-lactams and chloramphenicol [45]. In China, 60.22% of isolates from diarrheal animals (dogs, minks, cattle, fowl) were MDR, with 16.48% XDR. Resistance rates were high for ampicillin (59.09%), ciprofloxacin (57.39%), streptomycin (55.68%), doxycycline (63.64%), and tetracycline (55.12%) [30]. Egyptian isolates from diarrheal dogs showed 100% resistance to penicillin, amoxicillin, and trimethoprim-sulfamethoxazole, with variable resistance to other β-lactams and tetracycline [46]. Fish isolates also showed MDR and XDR to eight antibiotic classes, including β-lactams and polymyxins [79].

Widespread resistance in farm animals raises public health concerns regarding food safety. MDR rates were reported as 76.5% in chicken, 46% in pork, and 6% in beef isolates; chicken strains showed the highest resistance except for chloramphenicol and florfenicol [56]. Another study of 490 strains from meat found chicken isolates had highest resistance to doxycycline (83.33%), β-lactams (82.87%), aminoglycosides (89.81%), sulfonamides (91.67%), and quinolones (54.17%), with 14.9% PDR [50]. In meat and aquatic products, 91% were MDR, with high resistance to β-lactams, quinolones, chloramphenicol, and trimethoprim-sulfamethoxazole [49]. Interestingly, *P. mirabilis* from vegetables also showed multidrug resistance across β-lactams, quinolones, aminoglycosides, and tetracyclines, suggesting cross-contamination between animal and non-animal food sources [53].

In addition to genomic similarities, antibiotic resistance profiles of human- and animal-derived *P. mirabilis* strains also show notable similarities (see Figure 1). Marques et al. reported that *P. mirabilis* strains isolated from companion animals and human with UTIs in Portugal harboured common antibiotic resistance [45]. Both companion animal and human strains were sensitive to carbapenems like imipenem, meropenem, and ertapenem. Similarly, Yu et al. found higher resistance rates in poultry-derived *P. mirabilis* strains compared to human patient isolates for quinolones (ciprofloxacin) and penicillins (ampicillin), with poultry strains showing 48% (25/52) resistance to ciprofloxacin and 62% (32/52) resistance to ampicillin, compared to 35% (17/48) and 44% (21/48) in human isolates. Moreover, both groups exhibited two MDR profiles, suggesting the potential for cross-transmission of resistance genes [52]. These findings indicate a risk of resistance transmission between clinical and food sources.

Furthermore, similar antibiotic resistance patterns were observed between isolates from animals and those from animal-derived food (shown in Figure 1). Chinnam et al. found that 72 *P. mirabilis* strains (31.03%) from 232 animal and food sources were positive for β-lactamase production, including 60 strains confirmed to produce extended-spectrum β-lactamases (ESBLs), which were resistant to ceftazidime and cefotaxime but could be inhibited by β-lactamase inhibitors. About 42% of the sub-clusters contained strains from different hosts, indicating the potential for cross-contamination in slaughterhouse environments [31].

## 7. Antibiotic Resistance Genes Identified in *P. mirabilis*

Antibiotic resistance genes (ARGs) contribute significantly to the resistance phenotypes of *P. mirabilis* isolates and have been identified across various animal sources and related products (see Appendix A). For β-lactam antibiotics, resistance genes *bla_OXA-1_*, *bla_PSE_*, *bla_TEM_*, *bla_CTX-M_*, *bla_SHV_*, *bla_IMP_* and *bla_NDM_* have been found in isolates from farmed wildlife (fox, raccoon, and mink) and wildlife in natural environments (fruit bat, wild boar, gorilla, mandrill, African buffalo, sea lion, Eurasian goshawk and barn owl). resistance genes *bla_OXA-48_*, *bla_TEM_*, *bla_TEM-1_*, *bla_SHV-28_*, *bla_SHV-12_*, *bla_CTX-M-G1_*, *bla_CMY-1_*, *bla_CMY-2_*, andhave been found in isolates from wild mammals [19,20,21,22,23,27]. In farm animals such as pigs, chickens, and ducks, a wider range of resistance genes was detected, including *norA*, *acrB*, *bla_OXA_*, *bla_TEM_*, *bla_CTX-M_*, *bla_NDM_*, *bla_DHA_*, and *bla_KPC_* [31,32,34,35,36,37,73]. Isolates from companion animals (dogs and cats) carried *bla_OXA-1_*, *bla_TEM_*, *bla_CTX-M_*, and *bla_DHA_* [39,45,46], while foodborne isolates harbored *bla_CTX-M_*, *bla_OXA_*, *bla_DHA_*, *bla_CMY-2_*_,_ *bla_NDM_*, *bla_TEM_*, *bla_SHV_*, *bla_FOX_*, *bla_CIT_*, *bla_EBC_*, and *ble_MBL_* [31,49,53,56].

For quinolones antibiotics, resistance genes *qnrA* was detected in isolates from wild mammals [20]. Resistance genes *qnrA* and *qnrC* were detected in isolates from farm-raied wild animals [27], and additional genes such as *qnrS*, *parC*, *qnrD*, and *oqxA* were reported in farm animal isolates [32,34,35]. *qnrA* and *qnrD* were also found in isolates from companion animals [39,45,46], while only *qnrD* was identified in food-derived isolates [49,56]. Aminoglycosides resistance genes including *aac(6′)-Ib-cr*, *aadA*, *aadB*, *aphA6*, and *aaC2* were identified in farmed wild animal isolates [27], while only *aac(3)-II* was identified in wild mammals [20]. Farm animals carried *aac(6′)-Ib-cr*, *aph(3)-IIa*, *rmtB*, *aaC1*, and *aaC2* [32,34,35,73], while companion animals harboured *aphAI-IAB*, *aac(3′)-IV*, *aac(6′)-Ib*, and *aadA1* [39,45,46]. In foodborne isolates, *aac(6′)-Ib-cr*, *aph(4)-Ia*, *aadA1*, *aadA2*, *aac(3′)-Ia*, *aac(3)-IV*, and *aac(3)-IVa* were detected [49,53,56].

Tetracycline resistance genes such as *tetO*, *catI*, *tet(J)*, *tetA (48)*, *tetA*, *tetB*, *tet(C)*, and *tetM* were found in isolates from wildlife, farm animals, companion animals, and food sources [20,32,37,39,46,49,53,72,73]. Sulfonamide resistance genes (*sul1*, *sul2*, *sul3*, *dfrIa*) and chloramphenicol resistance genes (*floR*, *catB3*, *cml*, *cmlA*, *stcM*, *cat*, *cat1*, *cat2*) were widely present in isolates from farm animals (pigs, poultry) [32,35,37,72,73], companion animals (dogs, cats) [45,46], wild animals (foxes, raccoons, minks) [27], and food sources (chicken, pork, beef, vegetables, aquatic products) [49,53,56].

Resistance genes to macrolides, including *mphE*, *ermB*, *mefA*, and *mrsD*, were only reported in isolates from pigs and chickens [26,28]. The polymyxin resistance gene *mcr-1* and glycopeptide resistance gene *mecA* were exclusively found in chicken-derived isolates [36]. *fos* and *fosA3*, related to fosfomycin resistance, were identified in isolates from pigs and chickens, as well as food sources such as chicken, pork, and aquatic products [32,49,56].

Lincosamide resistance genes *cfr* and *lnu(F)* were detected by Ma et al. and Li et al. in food sources including chicken, pork, aquatic products, and vegetables. The same teams also identified the tigecycline resistance gene *tmexCD3-toprJ1* and the rifampin resistance gene *arr-3*, respectively [49,53]. In addition, the rifampin resistance gene *arr-3* and the disinfectant resistance gene *qacH* were found in pig-derived isolates [32]. The co-occurrence of *arr-3* and *qacH* may enhance resistance to rifampin and reduce susceptibility to common disinfectants, potentially compromising hygiene measures and increasing the risk of cross-contamination.

## 8. Perspectives

The emergence and spread of *P. mirabilis* across diverse animal hosts and food products, coupled with its multidrug resistance and virulence, underscore its growing importance as a zoonotic and foodborne pathogen. The widespread detection of clinically relevant ARGs such as *bla_NDM_*, *mcr-1*, and *tmexCD3-toprJ1* in isolates from farm animals, companion animals, wildlife, and animal-derived food highlight the urgent need for integrated surveillance systems. Global genomic analyses have revealed the transmission of antibiotic resistance genes (ARGs) among humans, animal sources, and the environment, highlighting their critical role within the One Health framework [82]. Antibiotic residues and drug-resistant bacteria present in urban domestic sewage and agricultural breeding wastes can enter the habitats of wild animals through environmental media such as soil and water bodies. In wildlife (e.g., wildboar and raptors) that have close contact with urban and agricultural areas, strains carrying the *bla_SHV-12_* gene have been isolated. This phenomenon indicates that the environmental pollution of drug resistance caused by human activities has extended to wild animal populations. Moreover, the identification of resistance genes in vectors such as houseflies and in contaminated slaughterhouse environments suggests overlooked transmission routes that warrant further investigation.

In terms of pathogenicity, *P. mirabilis* has been shown to cause similar pathological injuries in both animals and humans, suggesting a common bacterial pathogenic mechanism that is not host-specific. Pathogenicity can spread alongside the bacterium and its virulence genes, as horizontal transfer of these genes has been observed between isolates from captive pandas via mobile genetic elements [83]. Although numerous studies have reported the presence of virulence genes in *P. mirabilis* isolates from both animals and humans, comparative analyses of phylogenetic patterns and virulence at the genomic level between the two sources are still lacking. Notably, genomic studies have revealed a high degree of heterogeneity within *P. mirabilis* populations. Therefore, further investigation into the phylotypes of virulence and in-depth analysis of its pathogenic mechanisms is warranted to better understand its cross-host pathogenic potential.

Moving forward, a One Health approach should be emphasized, integrating data from human, animal, food, and environmental sectors to better understand the ecology and evolution of *P. mirabilis*. Molecular epidemiology tools such as whole-genome sequencing, resistome and virulome profiling, and comparative genomics will be invaluable for tracking its transmission and adaptation mechanisms. In addition, studies exploring biofilm formation, quorum sensing, and host–pathogen interactions may offer new targets for intervention. As *P. mirabilis* continues to adapt and spread under antibiotic selective pressure, coordinated international efforts are needed to mitigate its threat to public health and food safety.

## Figures and Tables

**Figure 1 microorganisms-13-02060-f001:**
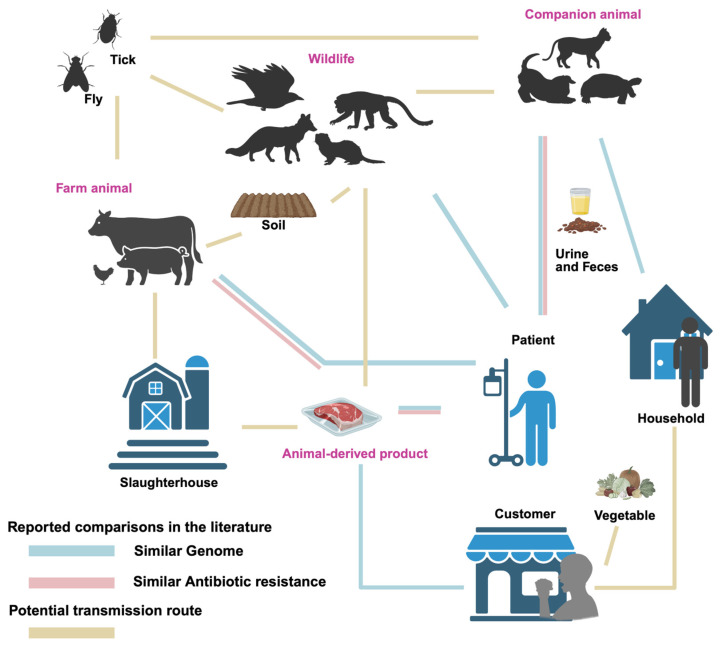
Transmission routes of *Proteus mirabilis* between animals, products, environments, and humans. Line for similar genome: Similarity of genomes between isolates from both sources has been reported. Line for similar antibiotic resistance: Similar patterns of antibiotic resistance have been reported. Line for potential route: Potential connection between sources, but not yet studied.

## Data Availability

No new data were created or analyzed in this study. Data sharing is not applicable to this article.

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
