# Peer review of "Zoonotic Risks of Proteus mirabilis: Detection, Pathogenicity, and Antibiotic Resistance in Animals and Animal-Derived Foods"

_microorganisms, 2025, doi:10.3390/microorganisms13092060_

Round 1
Reviewer 1 Report
Comments and Suggestions for Authors
- Table 1 is not cited in the text
- The use of italics for scientific names should be reviewed.
- The incorrect use of italics in text where it is not necessary should be reviewed.
- Figure 1 is not referenced in the text.
- Authors should explain or justify why there are no lines indicating potential routes between farm animals and patients, or include a fourth line indicating fully proven routes of infection or transmission. Alternatively, they should specify that the red and blue lines are proven routes, with specific distinctions between genomes and antimicrobial resistance.
- In Figure 1 the authors must define whether it is antimicrobial resistance or antibiotic resistance.
- The document addresses a widely documented topic, so the document does not represent anything new. It is recommended that you include some figure that may refer to genetic differences, which can be very helpful in enriching the content of the document.
Author Response
Dear reviewer,
We sincerely thank you for your patient and valuable suggestions on our manuscript. We have carefully addressed all your comments, and our detailed responses are provided below.
- Table 1 is not cited in the text.
Reply:Thank you for your kind reminder. In addition to the previous citation at line 62, a more appropriate citations have now been added, such as line 66, line 95, line 111, line 129.
- The use of italics for scientific names should be reviewed.
Reply: We have carefully reviewed and corrected the formatting of scientific names, ensuring they are italicized appropriately throughout the manuscript.
- The incorrect use of italics in text where it is not necessary should be reviewed.
Reply: We apologize for the formatting oversight. The incorrect use of italics has been reviewed and corrected in the revised manuscript.
- Figure 1 is not referenced in the text.
Reply: We apologize for the omission. Figure 1 has now been correctly referenced in the text at line 156, line 317, line 329.
- Authors should explain or justify why there are no lines indicating potential routes between farm animals and patients, or include a fourth line indicating fully proven routes of infection or transmission. Alternatively, they should specify that the red and blue lines are proven routes, with specific distinctions between genomes and antimicrobial resistance.
Reply: Thank you for your valuable suggestion. We have reviewed the relevant literature and found that a comparison between duck isolates and human isolates has been conducted based on a conserved gene. Accordingly, we have added a blue line in the figure to reflect this connection. In addition, the figure legend has been revised to provide a clearer explanation of the routes and distinctions represented.
- In Figure 1 the authors must define whether it is antimicrobial resistance or antibiotic resistance.
Reply: Thank you very much for the professional suggestion. Based on the referenced literature, only a limited number of non-antibiotic antimicrobial agents were mentioned. Therefore, we consider “antibiotic resistance” to be more appropriate in the context of this manuscript. Both the figure and the corresponding text have been updated accordingly.
- The document addresses a widely documented topic, so the document does not represent anything new. It is recommended that you include some figure that may refer to genetic differences, which can be very helpful in enriching the content of the document.
Reply: Thank you for your suggestion. In this review, our main objective was to highlight the connection between P. mirabilis in animals and humans based on reported data, with a focus on its zoonotic potential under the One Health framework. Although genetic dissimilarity was not the primary focus of this work, we have addressed relevant findings in the text to provide additional context.
Reviewer 2 Report
Comments and Suggestions for Authors
The paper is interesting but too long and too descriptive with too much examples and percentages that render the paper difficult to read. It has to be simplify and shortened focusing more on the virulence role of the bacterium and the demonstration of the sources and transmission ways. The pathogenic role of this bacterial species that is included in normal microbiota has to be clarify especially for the strains isolated from wildlife and healthy sources, please comment on this in the conclusions.
Author Response
Dear Reviewer,
Thank you very much for your careful review of our manuscript and your valuable suggestions. We have revised several sections to make the presentation of examples and data more concise and clear. This review primarily focuses on the connection of P. mirabilis between animals and humans; we gathered current reported evidence from the literature and summarized it in a single figure.
We agree that virulence is an important aspect to bridge the knowledge gap, so we included a section on pathogenicity. However, although many studies have analyzed pathogenicity and virulence genes, direct comparisons between animal and human isolates based on these factors are still lacking.
Following your professional suggestion, we have now added a discussion on the pathogenic role of P. mirabilis in the perspective section (lines 397–406).
Thank you again for your insightful comments.
The authors
Reviewer 3 Report
Comments and Suggestions for Authors
Dear Authors,
I have read your work thoroughly, this review will benefit the readers, since it's relates to the potential zoonotic risk of this commensal bacterium. I consider it is very useful also because it's presented the the genomic similarities between humans and animals, thus placed in the One Health nowadays concept. Please find some comments and suggestions I made to help improve this article:
- line 30 - instead of reference 2, which I don't consider it suitable here, should be placed the link of LPSN, that database is found under the form of a link
- line 43, line 158, line 199 etc - P. mirabilis should be italic, I suggest the be checked throughout the whole paper
- at the paragraph with wild animals (lines 63-89 and table 1) I think it could be added more data from different parts of the world, almost 80% of the data from the part is found only in China
- why the need to write all the paragraph of genomic similarity with italic? There are many "-" placed within the words, must be reviewed. Also, that subtitle should be written without "-"
- line 284, line 294 etc.- should be written susceptibility instead of sensitivity, is more accurate
Best regards,
The reviewer
Author Response
Dear Reviewer,
We sincerely thank you for your acknowledgment and valuable comments on our manuscript. Based on your suggestions and feedback, we have revised the manuscript accordingly. Our detailed responses are provided below.
Sincerely,
The Authors
line 43, line 158, line 199 etc - P. mirabilis should be italic, I suggest the be checked throughout the whole paper
Reply: The italic name has now been revised throughout the whole manuscript.
at the paragraph with wild animals (lines 63-89 and table 1) I think it could be added more data from different parts of the world, almost 80% of the data from the part is found only in China
Thank you for your kind reminder. We have now re-examined the literature and included additional data on wildlife from various countries.
why the need to write all the paragraph of genomic similarity with italic? There are many "-" placed within the words, must be reviewed. Also, that subtitle should be written without "-"
line 284, line 294 etc.- should be written susceptibility instead of sensitivity, is more accurate
Reply: We apologize for the formatting errors when adapting the text. The relevant sections have now been revised accordingly. Additionally, we reviewed and corrected the use of hyphens throughout the manuscript, minimizing their use where unnecessary. The term “animal-derived” has been retained as a standard compound adjective.
Round 2
Reviewer 1 Report
Comments and Suggestions for Authors
The authors covered the requested observations
Author Response
Thank you once again for your careful review and insightful feedback.
Reviewer 2 Report
Comments and Suggestions for Authors
I didnt received any specific response to my commentsto the first version , I only see some parts of the text in yellow and I suppose they have been modified but I dont know how. The text continous to be too long , full of too much examples of animals or material with P. mirabilis but the meaning of this presence is lacking. Is also lacking the part of the pathogenicity contribution of this bacterial species; there is not any demonstration that this bacteria can pass from animals or materials to humans causing infections. For these reasons the paper just describe the presence of P. mirabilis in animals and environment without analyzing the role of this species in human health
Author Response
As stated in our previous response, we have substantially revised the manuscript to streamline the examples and data presentation, reducing the length by approximately 1,000 words while retaining essential findings. The inclusion of these data, drawn from current literature, is critical as they form the evidence base for this review and are necessary to provide a comprehensive overview of P. mirabilis occurrence in animals and animal-derived foods.
With regard to pathogenic transmission, we acknowledge that no studies to date have provided direct proof of P. mirabilis transferring from animals or animal-derived materials to humans and subsequently causing infection—aside from a few documented food poisoning cases. This gap in direct evidence is precisely why our review is needed. By synthesizing reports on genomic similarity and antimicrobial resistance patterns between animal and human isolates, we highlight plausible transmission pathways that merit further investigation.
It is also important to clarify that our scope focuses on compiling and analyzing data from animal and animal-derived food sources, incorporating comparisons with human isolates only when strong evidence of similarity exists. In this way, the zoonotic potential of P. mirabilis is addressed in accordance with the currently available scientific evidence. In the revised version, we have also incorporated two newly published references to ensure the review is up to date.